# Effect of Electrode Morphology on Performance of Ionic Actuators Based on Vat Photopolymerized Membranes

**DOI:** 10.3390/membranes12111110

**Published:** 2022-11-07

**Authors:** Oleg S. Morozov, Anna V. Ivanchenko, Sergey S. Nechausov, Boris A. Bulgakov

**Affiliations:** Department of Chemistry, Lomonosov Moscow State University, 119991 Moscow, Russia

**Keywords:** electro-active polymer, ion exchange membrane, carbon nanotube, printing fabrication, additive manufacturing, vat photopolymerization

## Abstract

Bucky gel electrodes are composed of morphology-determining polyvinylidene difluoride (PVDF) filled with carbon nanotubes (CNT). The electrodes are commonly fabricated via the casting of a CNT dispersion containing PVDF and ionic liquid. In this study, several pore-forming additives such as polyethylene glycol (PEG), dibutyl phthalate (DBP), and the common ionic liquid BMIMBF4 were used to control the morphology of the bucky gel electrodes. The crystalline phase type and content of PVDF in the electrodes were determined by FT-IR and DSC, respectively. SEM revealed a sponge-like structure in the case of the use of BMIMBF4 and a spherulite structure if PEG and DBP were used as additives. A strong influence of morphology on the anisotropic increase in the volume of electrodes upon impregnation with electrolyte was observed. The PEG-based electrode elongated more than the others, while the BMIMBF4-based electrode thickened to a greater extent. Ionic actuators were fabricated to experimentally reveal the effect of electrode morphology on their electromechanical efficiency. A high-precision vat photopolymerization technique was used to fabricate identical ionic membranes and minimize their influence on the properties of the actuators. The electrodes were characterized by the same porosity and electrical capacitance, while the actuators differ significantly in performance. As a result, a simple method of using pore-forming additives made it possible to increase the maximum deformation of bucky gel ionic actuators by 1.5 times by changing the morphology of the electrodes.

## 1. Introduction

Soft material transductors exhibit flexible motion by changing the shape or volume, in contrast to conventional mechanical transducers such as electric motors, internal combustion engines, and pneumatic devices, in which motion is generated by changing the relative positions between their parts. One of the most promising technologies based on such materials is electromechanical transducers based on ionic electroactive polymers (*i*-EAP) [1,2,3]. Ionic EAP actuators are three-layer devices composed of an electrolyte membrane coated on both sides with electrode layers. Hydrated Nafion has become the most common electrolyte membrane, in part due to its availability and the fact that the first *i*-EAP devices were made with it [4,5]. These i-EAP actuators have shown great potential for applications as artificial muscles [3,6], underwater [7], bionic robots [8], and microelectromechanical systems (MEMS) [9,10]. A significant limitation for the further expansion of the scope of potential applications is the use of water or aqueous solutions as an electrolyte. Encapsulation makes it possible to fabricate stable actuators for long-term operation in air, although this approach does not solve the problem of the narrow electrochemical window of water [11,12,13]. Significant progress in increasing the stability of actuators has been achieved through the use of room-temperature ionic liquids [14,15,16,17]. Actuators based on non-volatile ionic liquids as electrolytes have revealed possibilities for applications in space [18,19,20]. On the other hand, ionic liquids have a relatively low conductivity, which decreases the actuation speed. Since the deformation of the actuator occurs due to the difference in volumes of migrating ions, it is possible to reduce the concentration of ions of the same type using ionomers for the fabrication of single-ion conducting membranes [21,22,23,24]. Another approach is to increase the effective surface area of electrodes. In 2003, research by Fukushima showed that highly entangled carbon nanotube (CNT) bundles fluff up in the presence of imidazolium salts, forming a physically crosslinked structure of ionic liquids called bucky gels [25]. High-performance ionic actuators based on bucky gel electrodes were described by Asaka [17,26,27]. Further modifications of electrodes were carried out either by replacing CNTs with other carbon materials such as carbide-derived carbon [28], graphitic carbon nitride [29], graphdiyne [30], and other carbon materials [31,32] or by introducing additives [33,34]. Since such actuators consist of three layers, there are problems with the reproducibility of the manufacturing technology as well as delamination of the layers. To overcome these difficulties, it was proposed to use additive manufacturing technologies [35,36,37,38]. Another interesting direction of research aimed at improving the efficiency of ionic actuators is the control of electrode morphology. Regardless of the mechanism of charge transfer, bending of the actuator occurs due to a change in the volume of the electrodes. In this case, changes in volume can occur unevenly along different axes. For example, the introduction of vertically aligned carbon nanotubes into Nafion led to a significant increase in lateral strain [39]. Previously, we have shown that the introduction of various pore-forming agents can effectively control the morphology of porous PVDF films [40,41]. It had also been shown that upon impregnation, the spherulitic structure of PVDF elongated to a greater extent, while the sponge-like structure thickened more.

Herein, we apply a similar approach to tune the morphology of bucky gel electrodes in order to increase their lateral deformation. We have shown that, regardless of the pore-forming additive used, the electrodes are characterized by equal porosity and electrical capacitance, while they differ significantly in structure. This difference in the morphologies provides for different elongations of the electrodes in ionic actuators when voltage is applied. To achieve high dimensional accuracy and reproducibility of the properties of actuators, a manufacturing method based on printing membranes via vat photopolymerization has been developed. The actuators were tested at 3 V, and a strong effect of morphology on the electromechanical performance was observed. Only the introduction of a pore-forming additive into a commercially available CNT dispersion makes it possible to increase by 1.5 times the maximal deformation of actuators. 

## 2. Materials and Methods

### 2.1. Materials

Single wall carbon nanotubes (SWCNT) as a TUBALL^TM^ dispersion contained 0.2 wt.% of nanotubes and 2 wt.% of PVDF in N-methyl-2-pyrrolidone and were purchased from OCSiAl (Russia). Polyethylene glycol with an average mass of 4000 (PEG), dibutyl phthalate (DBP), and N-methyl-2-pyrrolidone (NMP) were purchased from commercial sources (Acros, Aldrich) and were used without further purification. The compounds 1-butyl-3-methylimidazolium tetrafluoroborate (BMIMBF_4_) and 1-methyl-3-octylimidazolium tetrafluoroborate (OMIMBF_4_) were synthesized as described elsewhere [42].

### 2.2. Electrode Preparation

In a 100 mL beaker, 23.95 g of SWCNT dispersion was mixed with 23.42 g of NMP, and the calculated amount of additive is shown in Table 1. The mixtures were stirred at 1400 rpm for 2 h, poured into Petri dishes. Then the blends were cast in Petri dishes (Ø145 mm), and put into an oven with no forced convection preheated to 100 °C for 8 h. Dry electrode films were removed from the Petri dishes and weighted. The films were washed with acetonitrile to remove pore-forming additives. The procedure was repeated several times to achieve a constant weight for the samples.

### 2.3. Membrane Preparation

The ionogel membranes were 3D printed as a single layer with a thickness of 100 µm using an LCD 3D printer (Sonic 4K, Phrozen) with a LED ultraviolet light source at 405 nm and a light intensity of 5 mW/cm^2^. For printing, compositions based on the synthesized 1-alkyl-3-methylimidazolium ionic liquids, vinylpyrrolidone (VP), and triethylene glycol dimethacrylate (TEGDMA) were used (Table 2).

### 2.4. Actuator Fabrication

Rectangular samples (45 × 25 mm^2^) were cut from the washed electrodes and then soaked in a photopolymer composition to a constant mass. Impregnation was carried out in the dark to avoid premature polymerization. The impregnated electrode samples were wiped of an excessive photopolymer composition and cut with a blade into rectangles measuring (40.0 ± 0.5) × (5.0 ± 0.5) mm^2^. Then the printed membranes were sandwiched between the obtained electrodes. The sandwiched samples were placed on a steel plate, and a vacuum bag of polyimide film was assembled around the plate. A silicone hose was attached to the bag with adhesive tape in such a way that its one end was inside the bag. Next, the hose was connected to a vacuum pump, and the bag was vacuumized to a pressure below 1 mm Hg. The hose was sealed with a clamp and disconnected from the pump. The vacuum bag with the samples was placed in an oven heated to a temperature of 80 ℃ for 2 h. Then it was removed from the oven, allowed to cool to room temperature, and then the vacuum bag was disassembled. At the final step, the outer parts of the membranes were cut off with a blade along the perimeter of the electrodes to obtain actuators.

### 2.5. Electrodes Characterization

Differential scanning calorimetry was performed using a TA Instruments Q20 V24.11 calorimeter at a heating rate of 10 °C min^−1^ under an N_2_ atmosphere. A degree of crystallinity Xc was determined by:Xc (%) = ΔH_m_/ΔH_m_* ω_PVDF_ × 100%(1)
where ΔH_m_* = 104.7 J/g [43] is the melting enthalpy of a fully crystalline PVDF, ΔH_m_ is the experimentally obtained value of the thermal effect from DSC data, and ω_PVDF_ is the mass fraction of PVDF in an electrode film.

Fourier Transform Infrared (FT-IR) spectra were recorded on a Bruker Tensor-27 spectrophotometer in the range of 200–650 cm^−1^ using KBr pellets.

The electrode samples were weighed (m_0_) and placed into BMIMBF_4_ for 1 h at 60 °C. Then the electrodes were removed from the electrolyte and weighed in a saturated state (m_s_). The excess of electrolyte on the surface was soaked up with filter paper absorption. The procedure was repeated three times to achieve a constant uptake value. The porosity (P) was calculated by the following equation:
P = [(m_s_ − m_0_)/ρ_b_]/(m_0_/ρ_p_ + (m_s_ − m_0_)/ρ_b_) × 100%,(2)
where ρ_b_ and ρ_p_ are BMIMBF_4_ (1.22 g/cm^3^) and PVDF (1.77 g/cm^3^) densities, respectively.

Electrode capacitance was determined from galvanostatic charge-discharge data. The samples were sandwiched between symmetrical cells containing two glassy carbon electrodes at a current of 0.15 mA until the voltage reached 1000 mV. Then the current was reversed until the voltage reached 0 mV. Ten cycles were made for each sample. Capacitance was calculated by:(3)$$C\left(Fg^{-1}\right)=4\times \frac{I_{d}\left(A\right)\times \Delta t\left(s\right)}{\left(\Delta V-V_{IR-drop}\right)\left(V\right)\times m_{SWCNT}\left(g\right)}$$
where Δt is the discharge time, I_d_ is the discharge current, and ΔV is the voltage change. 

The electrodes’ and actuator’s surfaces and cross-sections were analyzed by scanning electron microscopy (SEM) in a TECCAN Vega 3 at 30 kV. Prior to analysis, sample surfaces were gold-coated to achieve better image contrast. The electrode films were fractured in liquid nitrogen for cross-section morphology observation. SEM images were obtained using backscattered electron (BSE) and secondary electron (SE) detectors.

### 2.6. Actuator Characterization

Ionic conductivities of actuators were calculated from electrochemical impedance spectroscopy data over the frequency range from 0.1 Hz to 1 MHz using a P-45X potentiostat/galvanostat equipped with an FRA-24M module (Electrochemical Instruments). The samples were sandwiched between symmetrical cells containing two glassy carbon electrodes at a constant potential of 10 mV to measure membrane impedance, Z (Ω). The thickness of each sample was measured with a micrometer at five different points, and the average value L (cm) was estimated for the calculation. The conductivity (σ, S/cm) was determined by the equation:(4)$$\sigma =\frac{L}{Z\times A_{1}}$$
where A_1_—area of the actuator.

Each actuator was tested at an alternating voltage with an amplitude of 3 V to assess the quality of the resulting devices. First, the bending shape of the device serves as an indicator of its quality; if it differs from the curve of a circle, this indicates an uneven distribution of properties over the actuator. Second, asymmetric bending in both directions indicates defects. Finally, an increase in current during testing can be associated with electrode short-circuiting, redox reactions, or compromising the integrity of the membrane. Devices that did not meet at least one of these criteria were considered defective and were not used for further testing.

Displacement measurements were performed using (40.0 ± 0.5) × (5.0 ± 0.5) mm^2^ size actuator strips. An actuator strip was connected to the glassy carbon electrodes. Constant and alternating triangle-wave voltages were applied to the actuator strip by a P-45X potentiostat/galvanostat equipped with an FRA-24M module (Electrochemical Instruments). The displacement was measured relative to graph paper. Constant voltage (+3 V) and alternating voltage (±3 V) with 0.008 Hz were used. The voltage and current were monitored simultaneously with the software ES8 for the potentiostat. The displacement was controlled by video registration. The actuator strip showed a bending motion when the voltage was applied. The strain is calculated from the displacement [44] by:(5)$$\varepsilon =\frac{2Dh}{L^{2}+D^{2}}\times 100\%$$
where D—displacement, L—free length, and h—thickness. 

According to the theory of bending of a three-layer bucky-gel actuator [26], the rigidity of the electrolyte layer has a slight effect on deformation of the actuator, and the generated force depends only on the Young’s modulus of the electrodes. Actuators of the same size were fabricated for testing. The blocking force was measured using analytical balances similar to the method described elsewhere [45]. The maximum generated stress (σ) during the actuation motion was calculated by:(6)$$\sigma =\frac{6Fl}{bh^{2}}$$
where F—generated force and l, b, and h—free length, width, and thickness of the sample, respectively.

## 3. Results

### 3.1. Effect of Additive on Electrode Properties

It was shown earlier in our laboratory that the morphology of PVDF membranes could be controlled when using various pore-forming additives in the manufacturing process [40,41]. Polyethylene glycol (PEG) and dibutyl phthalate (DBP) give the spherulitic polymer structure, while ionic liquid gives the sponge structure. We found that the PVDF membrane’s geometric dimensions changed in different ways when impregnated with the electrolyte (Figure 1). Thus, the samples made with PEG with an average mass of 4000 showed the greatest elongation, while the membranes made with BMIMBF4 mainly increased in thickness. The change in linear dimensions and the calculated change in the volume of films during impregnation are presented in Table 3.

The behavior described before could be related to the membrane’s morphology. When an electrolyte impregnates the sponge-structured polymer, compressed pores of the structure are stretched, and volume increase mainly occurs in thickness (Figure 2a). In the case of the spherulite structure, polymer spherulites can move relative to each other, so the expansion of the polymer structure during impregnation occurs more uniformly in all directions. (Figure 2b).

To investigate the morphology effect on actuation performance, three types of electrodes with different pore-forming additives were fabricated by the solution casting technique. The amount of each additive was calculated from its densities so that its content in the final electrode was 40 vol.%. The commercial dispersion of nanotubes is too thixotropic to be evenly distributed over the casting mold; therefore, it was diluted twice with NMP before solution preparation. After drying, the pore-forming additives were extracted, and then the dry films were immersed in BMIMBF_4_ and saturated in electrolyte until constant weight. Calculated porosity, electrolyte content values, and thicknesses of saturated films are presented in Table 4. The electrolyte uptake corresponded to the calculated additive volume, which indicates that the entire porosity is open.

An indirect parameter to evaluate the structure of the films is the relative content of the amorphous and crystalline phases of PVDF. The degree of crystallinity of PVDF was calculated from melting enthalpies obtained by DSC (Appendix A). With the growth of the degree of crystallinity, the density of PVDF increases, and therefore the porosity of the films should increase. However, all electrodes are characterized by approximately the same porosity, and a higher degree of crystallinity results in thinner films. This is probably due to the high adhesion of PVDF and additives; therefore, during crystallization, no additional free volume is formed, but the film becomes thinner. The composition of the PVDF crystalline phase could be determined from FT-IR spectra [46]. Analysis of the spectra showed the presence of the gamma phase in all samples (see Appendix A). β phase was presented in samples made with DBF and BMIMBF_4_ as additives. In any case, characteristic bands of the alpha phase were not detected.

The morphology of the electrode films was investigated using scanning electron microscopy. The surfaces and cross-sections of the electrodes are shown in Figure 3. Micrographs of the surfaces clearly show that the use of all additives except for BMIMBF_4_ leads to the formation of a spherulite structure in PVDF, and in the case of PEG4000, the size of the spherulites is the smallest. Cross-sectional images were obtained in two different ways. The films were cut with scissors, and they show that all electrodes, except for the one obtained with BMIMBF_4_, have a spherulite structure (Figure 3b). The cross sections of all films cracked in liquid nitrogen look similar (Figure 3c), and it is difficult to draw any conclusions about the structure of the inner part of the electrodes. For comparison, images of cross-sections of PVDF films obtained with the same additives are presented in Figure 3d.

Due to the different morphology of the electrodes, an anisotropic increase in volume would be observed, as in the case of PVDF membranes. To experimentally verify the change in geometry during impregnation, rectangular samples were cut from the electrode films and their linear dimensions were measured. The length and width of the films were measured with a caliper; the thickness was measured using an electron microscope, since a micrometer cannot provide sufficient measurement accuracy. The films were then impregnated in ionic liquid to achieve a constant weight, and the dimensions were measured again. With the same volume change, samples made using PEG4000 expanded mainly in length, while samples with DBF and IL expanded mainly in thickness (Table 5). This may affect the operation of actuators based on such electrodes. It is known that when voltage is applied, the ions in actuators are redistributed under an electric field. Migrated electrolyte ions introduce themselves into the pores of the electrode and change its volume. The deformation of the electrode caused by this process bends the entire actuator. To obtain greater bending, the electrodes of the actuator must stretch mainly along the length of the device. Thus, we assume that the use of electrodes prepared with PEG4000 can have a positive effect on the deformation.

### 3.2. Actuators

Ionogel membranes are manufactured according to the procedure described in our previous work [47,48]. Initially, it was planned to produce membrane samples based on different 1-R-3-methylimidazolium tetrafluoroborates, where R = ethyl, butyl, or octyl. During the membrane’s fabrication, it was found that it was not possible to produce a membrane from the composition based on 1-ethyl-3-methylimidazolium tetrafluoroborate. The membranes of an appropriate size for actuator manufacturing turned out to be too fragile to make actuators with them; the membranes crumbled when trying to separate them from the substrate. Hence, these membranes were not used in further work. Actuators based on BMIm and OMIm compositions were successfully manufactured and used for further research.

Ionic electrolyte membranes containing two ionic liquids were printed via the vat photopolymerization process. The electrodes were impregnated with the same photopolymer mixture and utilized for actuator preparation. When printing, the photopolymerization conversion does not exceed 50%, and further polymerization could be achieved by heating in the presence of a radical initiator. To produce an actuator, a printed membrane of the appropriate composition was sandwiched between previously impregnated photopolymer mixture electrodes. The electrodes and membrane surfaces were smeared with a photopolymer mixture for better adhesion. The samples thus obtained were placed in a vacuum bag and put in the oven to provide polymerization reactions (Figure 4).

This approach allowed for the formation of a chemical bond between the electrodes and membrane. The connection quality was checked using SEM (Figure 5 and Appendix A).

To obtain membrane conductivity, the impedance of actuators was measured (Appendix A). It can be seen from Table 6 that the conductivity obtained for the actuators is close to values obtained previously for polyelectrolyte membranes. Thereby, it can be concluded that the electrical properties of the membrane are preserved during the actuator’s manufacturing.

The actuators were tested for strain generated under a constant voltage of 3 V. The current dependence curves are shown in Figure 6. The shape of the curves suggests that the electrodes could be characterized as double-layer capacitors. 

It is known that the charge accumulated in the electrodes and, thus, their capacity, is proportional to the generated strain. Deviation from this pattern indicates the presence of additional effects inside the actuator. To find out the effect of the morphology of the electrodes on the electromechanical characteristics of the actuators, it was decided to compare these two parameters. Specific capacitances were calculated from the discharge time of the GCD measurement. Data on capacitances and maximum generated strain of actuators at 3 V are collated in Table 7. Peak-to-peak deformation is shown on Figure 7.

As can be seen, capacities close in value correspond to completely different deformations. Since the capacities are close, the accumulated charge must be the same in all cases and correspond to the same charge transfer and, consequently, the same volume change. We assumed that due to the anisotropic increase in volume of the electrodes caused by the different morphology, the difference in generated strain would be observed.

The mechanical stress was calculated from the generated force according to the equation for cantilever beam bending. Blocking forces for BMIm-based actuators could not be obtained due to the insufficient rigidity of the devices. The results for actuators based on OMIm are presented in Table 8.

According to the bending theory of a three-layer beam, the central layer has almost no effect on the bending stress of the actuator. The generated force is only determined by the properties of the outermost layers; in the case of actuators, the generated force is determined by the properties of the electrodes. As can be seen from the table, the sample-based electrodes obtained with BMIMBF4 show the highest blocking force; therefore, the electrodes have the highest rigidity. It is known that a polymer with a sponge-like structure has a higher rigidity than a polymer with a spherulitic one. Thus, it is possible to indirectly confirm the formation of a sponge-like electrode structure in the case of using BMIMBF4 as an additive.

Finally, the actuators were tested for long-term performance. The actuators based on OMIm membranes showed stable actuation for 5000 cycles at ±3 V at a frequency of 2.5 Hz (Appendix A). No obvious damage to actuators was detected, neither during the deformation tests nor after them.

## 4. Conclusions

Bucky gel porous electrodes based on PVDF, and single-walled carbon nanotubes were prepared using different pore-forming agents. SEM revealed a sponge-like structure in the case of ionic liquid (BMIMBF_4_) and a spherulite structure in the case of polyethylene glycol and dibutyl phthalate as additives. According to FT-IR spectra, mostly the γ -phase of PVDF was formed in electrodes obtained with PEG and DBP, and β-phase was observed in electrodes obtained with BMImBF4. The degree of crystallinity was measured by DSC. The porosity of the electrodes turned out to be practically the same, and the increase in linear dimensions upon impregnation with the ionic liquid occurred in different ways. The volume change upon impregnation for all electrodes was approximately equal. However, the PEG-based electrode showed the maximum elongation. Actuators were fabricated using ionic membranes prepared via vat photopolymerization. The actuators were tested under a constant and alternating voltage of 3 V to estimate maximum strain and generate blocking force. The actuator with electrodes obtained with PEG showed the greatest deformation (0.65%), while the one based on electrodes obtained with BMIMBF_4_ showed the highest mechanical stress (0.263 MPa). Thus, an influence of PVDF-based electrode morphology on ionic actuator performance was demonstrated, along with a simple technique for tuning the operational characteristics of such actuators.

## Figures and Tables

**Figure 1 membranes-12-01110-f001:**
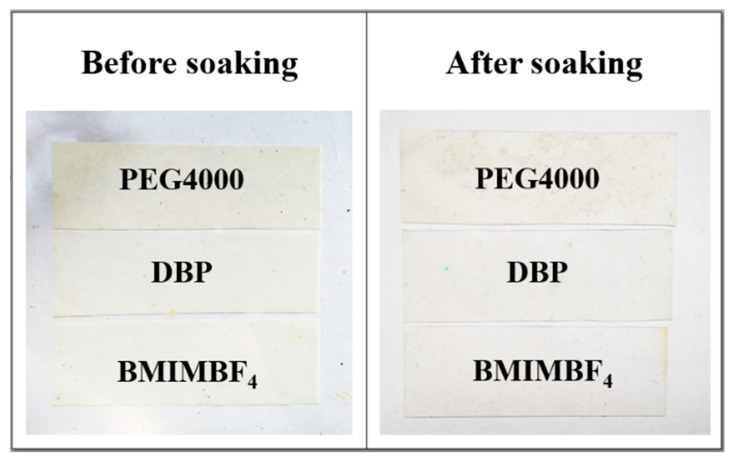
PVDF porous films washed from the pore-forming agent (left) and impregnated in BMIMBF4 (right).

**Figure 2 membranes-12-01110-f002:**
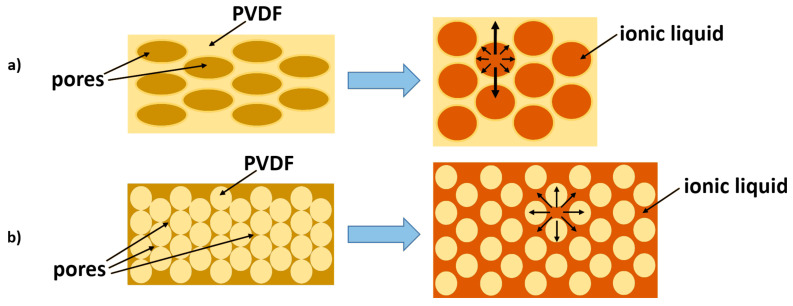
Impregnation of polymer membrane with: (**a**) sponge structure and (**b**) spherulite structure.

**Figure 3 membranes-12-01110-f003:**
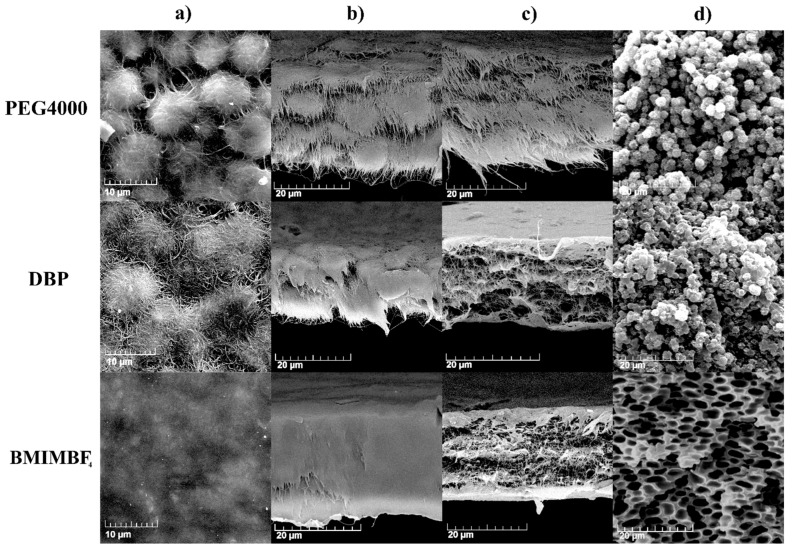
Micrographs of electrodes and PVDF membranes prepared with different pore-forming additives: (**a**) electrode surfaces; (**b**) cross-sections of the cut electrodes; (**c**) cross-sections of the torn electrodes; (**d**) cross-sections of the PVDF membranes.

**Figure 4 membranes-12-01110-f004:**
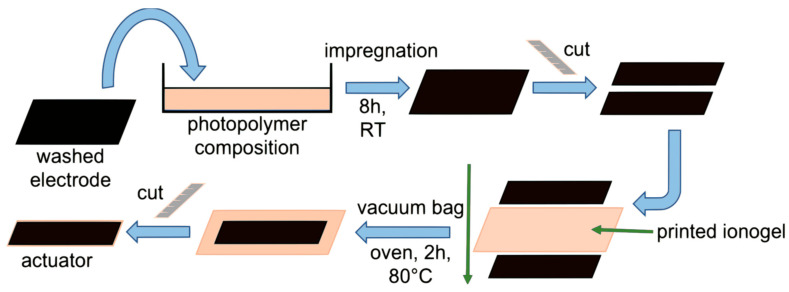
Manufacturing scheme for actuators.

**Figure 5 membranes-12-01110-f005:**
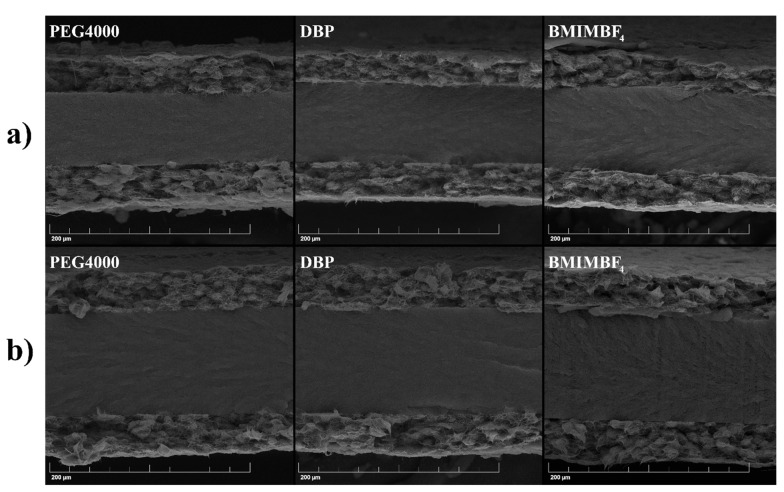
Microphotographs of cross-sections of actuators with: (**a**) BMIm membrane and (**b**) OMIm membrane.

**Figure 6 membranes-12-01110-f006:**
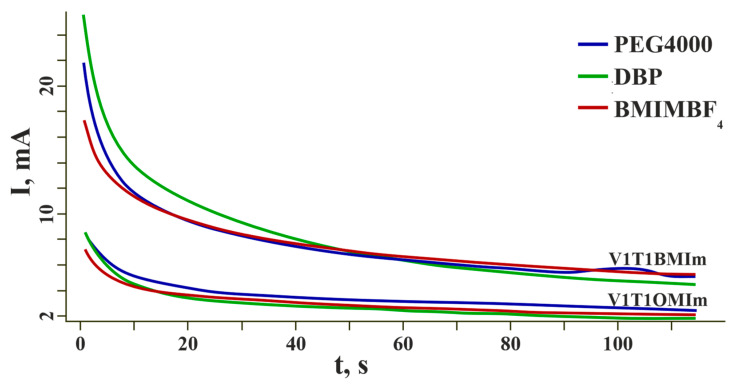
Time profiles of the generated current.

**Figure 7 membranes-12-01110-f007:**
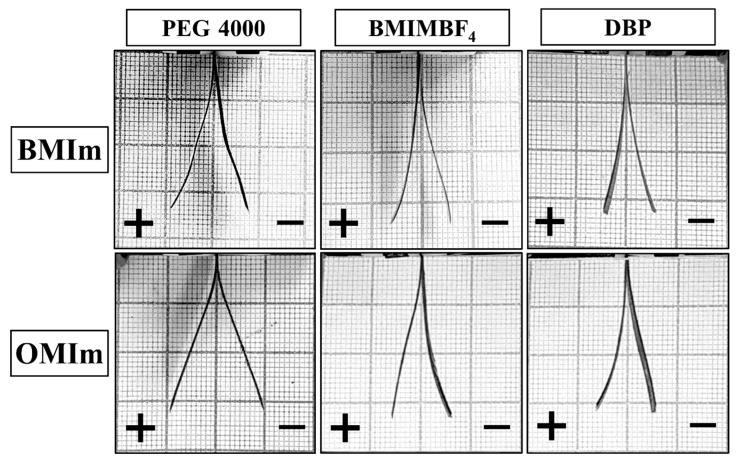
Peak-to-peak deformation.

**Table 1 membranes-12-01110-t001:** Pore-forming additives.

Additive	Density, g/cm^3^	Calculated Mass, mg
PEG	1.20	237
DBP	1.05	207
BMIMBF_4_	1.21	239

**Table 2 membranes-12-01110-t002:** Photopolymer compositions.

Abbreviation	Composition, % Mass.
BMIm	BMIMBF4/VP/TEGDMA 50/25/25
OMIm	OMIMBF4/VP/TEGDMA 50/25/25

**Table 3 membranes-12-01110-t003:** PVDF film parameters changing during impregnation.

Additive	Elongation, %	Thickening, %	Volume Increase, %
PEG4000	24.2	15	77.4
DBP	19.2	23	74.8
BMIMBF_4_	15.2	30	72.5

**Table 4 membranes-12-01110-t004:** Porosity, electrolyte uptake, thickness, DSC, and FT-IR data.

Additive	Porosity, vol. %	Electrolyte Uptake, vol. %	Thickness, μm	ΔH, J/g	Degree of Crystallinity, %	Crystalline Phases
PEG4000	41.3	41.1	30 ± 1	42.5	41	γ phase
DBP	40.4	40.0	33 ± 1	42.7	41	β + γ phases
BMIMBF_4_	39.6	39.7	31 ± 2	40.8	39	β + γ phases

**Table 5 membranes-12-01110-t005:** Electrode film parameters changing during impregnation.

Additive	Elongation, %	Thickening, %	Volume Increase, %
PEG4000	5	17	29
DBP	2	24	29
BMIMBF_4_	3	23	30

**Table 6 membranes-12-01110-t006:** Ionic conductivity of the membranes.

Abbreviation	σ_act_, mS/cm	σ, mS/cm [48]
BMIm	0.24	0.19
OMIm	0.10	0.11

**Table 7 membranes-12-01110-t007:** Electrode specific capacitances and actuation strains.

Additive	Capacitance, F/g_(CNT)_		Strain ±3 V, %	
	BMIm	OMIm	BMIm	OMIm
PEG4000	43.41 ± 0.23	43.43 ± 0.05	0.56	0.65
DBP	43.83 ± 0.64	42.99 ± 0.42	0.35	0.47
BMIMBF_4_	42.98 ± 0.23	43.46 ± 0.42	0.35	0.46

**Table 8 membranes-12-01110-t008:** The blocking force and maximum stress (σ) generated for OMIm actuators.

Additive	Blocking Force, mN	σ, MPa
PEG4000	0.196	0.126
DBP	0.235	0.170
BMIMBF_4_	0.274	0.263

## Data Availability

Not applicable.

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
