# Peer review of "Effect of Electrode Morphology on Performance of Ionic Actuators Based on Vat Photopolymerized Membranes"

_membranes, 2022, doi:10.3390/membranes12111110_

Round 1

Reviewer 1 Report

Electroactive polymers is a new promising development for sensors and actuators. The Nafion membranes used in the present work is widely employed in other applications such as fuel cells and microfluidics, so the described approach and the presented data might result in advancement in those areas as well. I must say I've enjoyed reading the manuscript and the findings surely got my attention, but the quality of presentation was distracting.

First of all I recommend stating the goal more clearly in the Abstract and in the Introduction and showing and if the results demonstrate that the goal was achieved in the Conclusion.

Second, please check the translation carefully. I've highlighted several issues below, but I am not a native English speaker so there might be some problems that eluded me. I strongly recommend consulting with an expert in English.

Please find the specific comments, both related and not related to translation, below:

 Please check how units are denoted. For example, you give [mSm] for conductivity, but while Siemens is denoted as См in Russian, it is denoted as S in English, so it should be mS. Also litres are denoted by the capital L in English to avoid confusion between small l, capital I and numeral 1 in some scripts.

Lines 90-91, "Mixtures were stirred at 1400 rpm for 2 (Ø 145 mm)" - I am not sure I follow, please clarify.

Line 113, "The samples obtained were placed in a vacuum bag" - please check the word order again, I am not a native English speaker either but it seems that it should be "the obtained samples <...>"

Line 117, "The differential scanning data were performed in" - I have doubts regarding the wording. First, please check if the word "calorimetry" was indeed not intended. Second, the word "performed" seems off with the subject "data".

Line 121, "where ΔHm* = 104.7 J/g []," - looks like a missing citation to me.

Please clarify why the porosity was measured in a way it was measured. For now or is not clear why the membranes were separated from the electrodes in process and if they were reapplied between the repeated measurements.

Line 196, "Figure 1. This is a figure. Schemes follow the same formatting." - seems like a placeholder text to me.

Please clarify the variables in the top row of table 3.

Please add more details on how the transition shown in figure 4 at the stage of heating under vacuum occurs.

Caption to figure 5 - I assume you've meant micrographs, not microphotographs, since "microphotograph" is a microscopic scale photograph.

Table 7 - why BMIm and OMIm in some places and BMIM and OMIM in others?

Line 300 onwards, "As can be seen, capacities close in value correspond to completely different deformations. Since the capacities are close, the accumulated charge must be the same in all cases and correspond to the same charge transfer and, consequently, the same volume change. We assumed that due to an anisotropic increase in volume of the electrodes caused by the different morphology, the difference in generated strain would be observed." - I like this observation and conclusion very much and if I were you I would address it in the Abstract.

***

Best wishes.

Reviewer 2 Report

The caption and axis labels in figure 6 can be improved.

The peak to peak deformation images need some scale bars for the readers.

Author Response

The caption and axis labels in figure 6 can be improved.

We have revised the drawing 6, as well as the drawings in additional materials. The axes should now be clearly visible.

The peak to peak deformation images need some scale bars for the readers.

The photographs of the actuators bending were taken against the background of millimeter paper, we considered it redundant to add a scale to these photographs.

Reviewer 3 Report

Dear authors,

Thank you for your interesting manuscript. I believe it can be accepted for publication after its revision. I have the following comments:

1. The annotation contains many abbreviations, but only two have been deciphered.

2. Line 121: «where ΔHm* = 104.7 J/g []». Missing reference.

3. Line 238: You probably meant Figure 3d.

4. Very small font size on the axes in Figure 6 and in Figures S5, S6, S7 in Supplementary.

5. «Figure 1. This is a figure. Schemes follow the same formatting.» The name of the picture is from the template

6. Supplementary information has more authors than the article.

7. I suggest the authors to add a conclusion section to the article in which to summarize the obtained results and compare the obtained characteristics of actuators with other types of actuators based on ionic electroactive polymers.

Author Response

Thank you for your interesting manuscript. I believe it can be accepted for publication after its revision. I have the following comments:

1. The annotation contains many abbreviations, but only two have been deciphered.

We have completely rewritten the abstract to make the goals and objectives of the work clearer for a wide range of readers. We revised the last paragraph of the introduction and added conclusions. Now the abstract reflects the main goal of the work and the results without technical details, the conclusions, on the contrary, briefly describe the specific results of the article.

2. Line 121: «where ΔHm* = 104.7 J/g []». Missing reference.

 The reference has been added.

3. Line 238: You probably meant Figure 3d.

Corrected

4. Very small font size on the axes in Figure 6 and in Figures S5, S6, S7 in Supplementary.

We have reworked the Figure 6, as well as the Figures in supplementary materials. The axes should now be clearly visible.

5. «Figure 1. This is a figure. Schemes follow the same formatting.» The name of the picture is from the template

We've added a caption to the Figure 1

6. Supplementary information has more authors than the article.

Corrected

7. I suggest the authors to add a conclusion section to the article in which to summarize the obtained results and compare the obtained characteristics of actuators with other types of actuators based on ionic electroactive polymers.
We added Conclusion section. 
We agree that comparison with known results always gives a better understanding of the place of work in the context of modern science. However, in this work, the focus is on model systems: the electrodes contain a fairly low amount of carbon nanotubes, and the choice of method for fabricating membranes and actuators is rather dictated by accuracy and reproducibility. We plan to apply the discovered regularities to the fabrication of highly efficient ionic actuators. And we plan to compare them with other actuators in the next paper.

Round 2

Reviewer 1 Report

Dear authors, I thank you for the revisions made. Very interesting and important changes were made in syntax and in structure. Please beware of typos that look like legitimate words (wight in line 96).

I am unhappy with how same tables are formatted (e.g. 3rd and 5th, line breaks at the headings in the narrow tables), but they would different in the final online version anyway.

Best regards.

Author Response

Dear authors, I thank you for the revisions made. Very interesting and important changes were made in syntax and in structure. Please beware of typos that look like legitimate words (wight in line 96).

Thanks for pointing out this typo. We rechecked the entire text and corrected some typos and grammatical errors.

I am unhappy with how same tables are formatted (e.g. 3rd and 5th, line breaks at the headings in the narrow tables), but they would different in the final online version anyway.

Corrected. We believe that at the stage of final editing of the article we will correct all such shortcomings, including word wraps and splitting tables into two pages.